# Current Guidelines for Diagnosing PCOS

**DOI:** 10.3390/diagnostics13061113

**Published:** 2023-03-15

**Authors:** Jacob P. Christ, Marcelle I. Cedars

**Affiliations:** Department of Obstetrics, Gynecology and Reproductive Sciences, University of California, San Francisco, CA 94143, USA

**Keywords:** polycystic ovary syndrome, diagnosis, Rotterdam criteria

## Abstract

Polycystic ovary syndrome (PCOS) is the most common endocrine disorder of reproductive-aged women. Much of the confusion surrounding PCOS diagnosis stems from the broad heterogeneity of symptomology experienced by women with PCOS. The diverse features of the syndrome have led to a number of diagnostic criteria over the years. This manuscript describes each of the current composite criteria and individually breaks down each component. The importance of accurate diagnosis for both clinical care and research is emphasized.

## 1. Introduction

Polycystic ovary syndrome (PCOS) is the most common endocrine disorder of reproductive-aged women [1]. Despite causing over USD 8 billion in healthcare costs in the United States, accurate US population-based estimates of incidence and prevalence are lacking and as many as 75% of patients with PCOS are unidentified in clinical practice [2]. Much of the confusion surrounding PCOS diagnosis stems from the broad heterogeneity of symptomology experienced by women with PCOS.

This syndrome is predominantly characterized by irregular menstrual cycles, hyperandrogenism, and characteristic findings on pelvic ultrasound [3,4]. Women with PCOS also frequently suffer from metabolic dysfunction, obesity, infertility and are at an increased risk of pregnancy complications and long-term cardiovascular disease [5,6,7,8,9,10]. There exists, however, significant heterogeneity among the phenotypic expressions of PCOS and disease sequelae may vary across a woman’s lifespan [11,12]. Furthermore, while there are around 30 genes associated with the development of PCOS [13,14,15], pathogenesis of this disease is complex, multi-factorial and not fully elucidated, thus diagnosis relies on identifying features of the syndrome following exclusion of known disorders affecting ovulation or hyperandrogenism.

The diverse and varying features associated with this syndrome have resulted in the proposal of several diagnostic criteria for this disease dating back to 1935, when first described by Stein and Leventhal [16]. Ideal diagnostic criteria will have optimized sensitivity and specificity, ensuring confidence in the criteria’s ability to identify affected and unaffected individuals [17]. The development of such a definition, however, requires agreement on a gold standard for disease classification, assumes technology used for diagnosis will not change, and relies on accurate evaluation of criteria components. Unfortunately, diagnosis of PCOS has been challenged by all three of these pillars. This review aims to highlight the evolution of the diagnostic criteria for PCOS over time and provide a summary of the most recent criteria proposed for the diagnosis of this syndrome.

## 2. Diagnostic Criteria for PCOS

In 1935, Stein and Leventhal first characterized what is now known as PCOS among a case series of seven women with a combination of hirsutism, obesity, amenorrhea and bilateral enlarged polycystic-appearing ovaries on surgical and pathologic evaluation [16]. Since then, several diagnostic criteria have been proposed which variably include a combination of oligo-amenorrhea, hyperandrogenism and changes in ovarian morphology, as now assessed by pelvic ultrasonography. 

In 1990, the first attempt to produce a clinical definition of PCOS was completed by the National Institute of Child Health and Human Development, in which PCOS was defined by the presence of both clinical and/or biochemical signs of hyperandrogenism and oligo- or chronic anovulation [18]. Ultrasonographic evidence of polycystic ovaries was reported as suggestive of PCOS, but not necessarily diagnostic, which conflicted with the leading practice in the United Kingdom and much of Europe at the time, whereby polycystic ovaries on ultrasound were viewed as the “defining feature of PCOS” [19]. This debate continued until 2003, when 27 PCOS experts met in Rotterdam, the Netherlands, at a conference sponsored by both the European Society of Human Reproduction (ESHRE) and American Society for Reproductive Medicine (ASRM), and produced a joint consensus statement commonly known as the “Rotterdam Criteria” [3,4]. These criteria broadened the phenotypic expression of PCOS to include any two out of the three key characteristics of PCOS: oligo-amenorrhea, hyperandrogenism, and polycystic-appearing ovarian morphology on ultrasonography. In doing so, the prevalence of PCOS, in some studies, increased as much as three times compared to diagnosis using the 1990 NIH criteria [20]. Furthermore, the use of these criteria allowed for the diagnosis of PCOS without hyperandrogenism, which had previously been viewed as the primary defect by the 1990 NIH criteria. 

Since the 2003 Rotterdam criteria, all proposed criteria have included ovarian morphology with varying degrees of importance. In 2006, the Androgen Excess Society (AES) again made hyperandrogenism central to the diagnosis of PCOS, while affirming the relevance of ovarian morphology in the diagnosis of this syndrome [21]. The AES guidelines, required the presence of hirsutism and/or biochemical hyperandrogenism, as well as either oligo-anovulation and/or polycystic-appearing ovarian morphology (PCOM) for the diagnosis of PCOS [21]. Thus, the “most mild” phenotype of PCOS (oligo-anovulatory women with polycystic ovarian morphology and without hyperandrogenemia) was excluded. 

The presence of multiple classification systems resulted in clinical confusion and was viewed as delaying scientific progress in our understanding of PCOS. Thus, in 2012, the NIH held an evidence-based methodology workshop on PCOS, in which experts on PCOS again recommended use of the broader 2003 Rotterdam criteria, while specifically identifying sub-phenotypes within these criteria of (1) androgen excess and ovulatory dysfunction, (2) androgen excess and PCOM, (3) ovulatory dysfunction and PCOM, and (4) androgen excess, ovulatory dysfunction and PCOM [22]. The Rotterdam criteria continues to be the most widely used and accepted criteria for PCOS and were once again unanimously supported in the 2018 International Evidence-Based Guideline for the Assessment and Management of PCOS [23]. The remaining sections will, thus, focus on the definition of each of the sub-components of the 2003 Rotterdam Criteria: hyperandrogenism, oligo-anovulation and PCOM. 

### Current Recommendations

It is recommended to use the modified Rotterdam criteria, (see Table 1) in which PCOS may be diagnosed if any two of the following are present: (1) clinical or biochemical hyperandrogenism, (2) evidence of oligo-anovulation, (3) polycystic appearing-ovarian morphology on ultrasound, with exclusion of other relevant disorders.

## 3. Androgen Excess

### 3.1. Hyperandrogenemia

In women, androgens normally function to support bone density, muscle mass and sexual function. Biosynthesis of androgens occurs 25% from the ovaries, 25% from the adrenal gland and 50% from peripheral tissues, and takes the form of testosterone, dihydrotestosterone, dehydroepiandrosterone sulfate (DHEA-S), dehydroepiandrosterone (DHEA), and androstenedione (ANSD) [24,25]. Testosterone is primarily produced by the ovaries, ANSD is produced evenly by the ovaries and adrenal gland, and DHEAS is produced exclusively by the adrenal gland [24,25].

The majority of oligo-amenorrheic patients with PCOS also have biochemical hyperandrogenemia [21,26]. The ovaries are the primary source of hyperandrogenism in patients with PCOS. Accordingly, testosterone, predominantly in the free form unbound to sex-hormone-binding globulin (SHBG), is most frequently elevated in these patients and is the most sensitive marker for diagnosis [27]. Obesity and hyperinsulinemia are associated with reduced SHBG, and thus levels are typically lower in patients with PCOS [8]. Up to 89% of patients with PCOS and hyperandrogenemia have been found to have elevated levels of free testosterone, while 49 to 80% of patients have been found to have elevated total testosterone [28,29,30], ANSD has been found to be elevated in up to 88% of patients with PCOS and may increase the number identified as having hyperandrogenemia by about 10%. DHEAS is elevated in 25–35% of patients with this syndrome and may be the sole abnormality in circulating androgens in about 10% of patients [21,26,31].

Measurement of serum androgens in women is challenging due to several considerations. Commonly available assays for the measurement of androgens are often unreliable at the lower limits seen in women. Additionally, the concentration of testosterone varies throughout the day and other similar steroids can cause assay interference [32]. Because of these issues, it is recommended that direct assays (those performed on whole serum) be avoided and assays after extraction and chromatography, followed either by mass-spectrometry or immunoassay, be used [23,32]. Free testosterone represents only 1–3% of all testosterone, thus highly precise assays are required for measurement. Equilibrium dialysis is considered the gold standard for measuring free testosterone, however this technique is relatively expensive and requires technical expertise [32]. Calculated estimations of free testosterone have thus been proposed, including free androgen index (the quotient testosterone/SHBG × 100] and calculated bioavailable testosterone (the concentration of testosterone that is free and weakly bound to albumin). These methods have been found to be relatively reliable markers in the diagnosis of PCOS [27]. Thus, calculated measures of free testosterone, or high quality assays for the measurement of total and free testosterone are recommended as the primary marker of hyperandrogenemia in patients with PCOS [23].

### 3.2. Current Recommendation

Biochemical hyperandrogenism should be defined by elevated total or free testosterone, as measured by high-quality assays such as liquid chromatography mass spectrometry and extraction/chromatography immunoassay. Calculated free testosterone, free androgen index or bioavailable testosterone may also be used to assess biochemical hyperandrogenism in the diagnosis of PCOS.

ANSD and DHEAS could be considered if total or free testosterone are not elevated.

Interpretation of androgen levels should be guided by the reference ranges of the laboratory used.

### 3.3. Clinical Hyperandrogenism

The clinical manifestations of elevated androgen levels in women include hirsutism, acne and female pattern hair loss (formerly referred to as androgenic alopecia). Hirsutism, excessive terminal hair growth in a male pattern distribution, is common in patients with PCOS, affecting 60–70% of people [33]. The degree of hair growth is typically quantified with the Modified Ferriman–Gallwey (MFG) scoring system, in which terminal hair growth, on a scale from 0 to 4, at nine different anatomic sites, is scored and scores are summed [34,35]. Different MFG score thresholds have been proposed to diagnose hyperandrogenism ranging from ≥3 to ≥8 [33,36,37,38]. Unfortunately, there remains interobserver variability despite the use of the MFG scoring system [39], as scoring may be limited by a patient’s use of personal hair removal treatments [40], and there is significant ethnic variability on MFG scoring [38]. Despite these limitations, MFG scoring, when completed by a trained provider, remains the gold standard for assessing clinical hyperandrogenism [23,38]. What threshold to use remains in question, as the most recent guideline recommend a threshold of ≥4 to 6, depending on ethnicity, which may result in the overdiagnosis of this condition as compared to the traditionally used threshold of ≥8 [23,41].

Female pattern hair loss and acne are common complaints among patients with PCOS [42,43]. While both have been associated with biochemical hyperandrogenism, only one third of women with female pattern hair loss has elevated androgen levels [44,45,46,47]. Female pattern hair loss can be assessed with the Ludwig visual scale, while there is no universally accepted visual assessment for evaluating acne [23]. Furthermore, data are lacking regarding the reliability of these features for the diagnosis of PCOS. Providers should, thus, be adept at managing these conditions when treating patients with PCOS, however, inclusion of these features in the diagnosis of this syndrome is not currently recommended.

### 3.4. Current Recommendations

Clinical hyperandrogenism should be evaluated by a trained provider and be quantified using the modified Ferriman–Gallwey score. The threshold used for “abnormal” may vary based on patient population from ≥4 to ≥8.

While acne and female pattern hair loss are common complaints among patients with PCOS, currently, data do not support their use as reliable diagnostic markers for PCOS.

## 4. Irregular Cycles and Ovulatory Dysfunction

The average menstrual cycle in adults lasts 28 days, with a normal range of 21–35 days, with typically a relatively constant luteal phase lasting 14 days, and a more variable follicular phase length [48]. Even among those with ovulatory cycles, however, there can be significant heterogeneity between individuals and within a single individual in overall cycle, follicular phase, and luteal phase length [49]. Generally, the presence of regular monthly menses occurring within this normal range can be used as a surrogate marker of ovulatory function [50]. It is not uncommon, however, even among women with regular 28-day cycles to have one or more anovulatory cycles in a year [51]. Based on these findings, the most recent guidelines recommend the use of irregular menstrual cycles as a marker for ovulatory dysfunction [23]. Menstrual dysfunction among patients with PCOS is typically characterized by oligo-amenorrhea (cycles > 35 days apart or <8 cycles per year). Polymenorrhea (cycles occurring < 21 days apart) is relatively uncommon among patients with PCOS [52], however this feature has been included in some guidelines for the diagnosis of PCOS [23,53]. Regardless, if there is a strong clinical suspicion for PCOS in a patient with regular menstrual cycles, polymenorrhea or an unclear menstrual pattern, additional assessment of ovulation should be confirmed with serum progesterone, given the possibility of ovulatory dysfunction without classic oligo-amenorrhea [23]. While out of the scope of this review, it is important to note that this recommendation only applies to adults. Irregular menses are normal within 1 year post-menarche, and in 1 to 3 years, post-menarche irregular menses should be defined as <21 or >45 days apart [54].

### Current Recommendation

Oligo-amenorrhea (cycles > 35 days apart or <8 cycles per year) may be used as a marker for ovulatory dysfunction to diagnose PCOS.

Ovulation can be confirmed in those with uncertain menstrual history with serum progesterone evaluation or luteinizing hormone testing.

## 5. Ovarian Morphology

Originally described by Stein and Leventhal in 1935 on surgical and pathologic examination as bilaterally enlarged, polycystic-appearing ovaries, polycystic-appearing ovarian morphology has continued to be a key component in the diagnosis of PCOS [16]. Since the 1980s, ultrasonography has allowed for non-invasive assessment of ovarian morphology [55]. The first set of most widely adopted criteria for PCOM, suggested by Adams et al. in 1985, defined this feature as 10 or more follicles 2–8 mm in size in one cross-section of the ovary on transabdominal ultrasonography [56]. Since then, transabdominal ultrasonography has largely been replaced by the higher resolution endo-vaginal approach. The first group to use receiver operator characteristic (ROC) curves to develop thresholds for PCOM was Jonard et al. in 2003, in which a follicle number per ovary threshold of 12 or more, measuring 2–9 mm in diameter (mean of both ovaries), had 75% sensitivity and 99% specificity in the diagnosis of PCOS [57]. The 2003 Rotterdam criteria based their recommendation for PCOM on this study and recommended PCOM to be defined as either 12 or more follicles measuring 2–9 mm in diameter or an ovarian volume > 10 cm^3^ for either ovary [58].

Several other markers of PCOM had previously been described. The classic description of PCOM included the appearance of an increased number of follicles 2–9 mm in size, arranged in a peripheral distribution (appearing as a “string of pearls”), around a bright echo dense stroma. However markers of the stromal area, stromal echogenicity and follicular distribution have not been found to have significant predictive power in the diagnosis of PCOS when used alone, and add little when combined with follicle number and/or ovarian volume [59]. As such, these features were excluded from the definition of PCOM in 2003 and by all major criteria since [58].

Over the past decade, much debate has ensued regarding the appropriate thresholds to be used in the definition of PCOS. Using the proposed thresholds for PCOM by the 2003 Rotterdam criteria, 30–50% of normo-androgenic, ovulatory women would meet criteria for PCOM [60,61,62]. This led many to conclude the thresholds for PCOM needed to be revised and studies since have presented varying proposed thresholds for PCOM [62,63,64,65,66,67]. Much variation between reported follicle counts can be explained by changes in ultrasound technology over time, in which increased transducer frequency (≥8 MHz) allows for improved detection of antral follicles on ultrasound, and thus elevated threshold levels [68]. Furthermore, some earlier studies excluded otherwise healthy patients with PCOM from control groups, resulting in lower cut-offs of the follicle number [67,69] as compared to those who included these patients [62]. In 2014, a task force report on the definition and significance of PCOM was produced by the Androgen Excess and PCOS (AEPCOS) society, in which all available literature on this subject was compiled and an updated recommendation for PCOM was provided. These guidelines recommended increasing the threshold to ≥25 follicles per ovary and/or an ovarian volume threshold of ≥10 cm^3^, based on transvaginal ultrasound with a transducer frequency of 8 MHz or greater. This increased threshold, however, has since been challenged, as it results in the exclusion of a large group of oligo-anovulatory women, provides limited added information on the degree of hyperandrogenism, and may exclude a group of women still at increased risk of metabolic dysfunction [70,71]. Most recently, a slightly reduced follicle number threshold has been proposed by the 2018 International Evidence Based Guidelines for the Assessment and Management of PCOS, at ≥20 follicles per ovary and/or an ovarian volume of ≥10 cm^3^ [23]. Whether use of different follicle number thresholds has true clinical relevance outside of providing a diagnostic label is debatable, as the degree of hyperandrogenemia better predicts metabolic risk and has more clinical relevance (in addition to oligo-anovulation) than ovarian morphology for most patients with PCOS [11]. Likely, ultrasound criteria will continue to evolve as technology improves and new, more reliable criteria continue to be developed.

Given the large degree of heterogeneity with respect to ultrasound assessment in PCOS, much interest has been placed on anti-Mullerian hormone (AMH) as a surrogate marker for ovarian morphology. AMH is a polypeptide secreted by granulosa cells of the preantral and small antral ovarian follicles [72]. Levels of AMH are significantly higher among those with PCOS compared to those without PCOS [73], and many have attempted to assess the diagnostic accuracy of this hormone for PCOS and PCOM [36,38,43,44,45,46,47,48,49,50,51,52,53,54,55,56]. While results are promising, there is significant heterogeneity between study methodologies and proposed diagnostic thresholds, and standardization of AMH measurement is still needed to ensure inter-assay accuracy [74]. Because of these limitations, most recent guidelines do not recommend the use of AMH levels as an alternative for the detection of PCOM or as a single test for the diagnosis of PCOS [23].

Follicle number and AMH are also known to decline over the lifespan in those with and without PCOS [75,76,77], which has led some to propose the need for age-specific criteria for PCOS diagnosis. Age-stratified thresholds for AMH have been found to improve the AMH predictive performance for the diagnosis PCOS compared to a single non-age-adjusted threshold [78,79,80]. Similarly, decreasing thresholds for follicle number and ovarian volume, with advanced age, have been suggested for the diagnosis of PCOS [81]. Age-specific thresholds have not yet been widely adopted, however, recognition of changes in these features over time is important, especially when evaluating older patients for PCOS.

### Current Recommendation

PCOM should be defined as either ≥20 follicles per ovary and/or an ovarian volume of ≥10 cm^3^ on either ovary, using newer transvaginal ultrasound technology with a transducer frequency of 8 MHZ or more.

At this time, AMH is not recommended as an alternative marker for PCOM and should not be used as a single test for the diagnosis of PCOS.

## 6. Conclusions

PCOS remains a clinical diagnosis, following the Rotterdam Criteria, requiring two of the three symptoms as follows: oligo-anovulation, hyperandrogenism and/or polycystic ovarian morphology (PCOM). While updates have occurred in the criterion utilized for PCOM, and there is a call for improvements and standardization, in the testosterone assay, the controversy since the initial publication of these criteria continues. As noted, acceptance of the Rotterdam Criteria for the diagnosis increased the prevalence of PCOS in the population. Perhaps, more importantly, the variability between the, now, multiple phenotypes of PCOS, compromised goals of better classification: improving research regarding underlying pathophysiology and risks of the diagnosis, and treatment recommendations for individual patients. At a minimum, studies should be very specific regarding the phenotype under investigation, rather than the broader “PCOS as defined by Rotterdam”.

Diagnosis of PCOS should not be given lightly. Receiving a diagnosis of this syndrome is associated with significant psychological distress [82], reduced well-being, depression, and fears about future health and fertility [83]. From the patient’s perspective, the vast majority receiving a diagnosis feel that they either receive no information about the diagnosis or receive inadequate information [84]. Furthermore, diagnosis can be delayed by two or more years for approximately a quarter of women with PCOS [82]. Whether the burden of receiving a PCOS diagnosis is a result of the diagnosis process, or is instead due to PCOS itself, is not completely clear [85]. Regardless, considering the importance of this syndrome, women are owed a timely and appropriate diagnosis. Given the continued debate over criteria, and the often inadequate clinical care using the current framework for diagnosis of PCOS, it begs the question: after almost 20 years, is it time to revisit this diagnosis?

## Figures and Tables

**Table 1 diagnostics-13-01113-t001:** Features of the diagnosis of PCOS.

Feature	Recommended Diagnosis	Considerations
Biochemical Hyperandrogenism	Elevated total or free testosterone, or calculated indices of free testosterone (FAI, BioT). DHEAS and ANSD can be consdered	High-quality assays should be used for the evaluation of analytes
Clinical Hyperandrogenism	A modified Ferriman–Gallwey score of ≥4 to ≥8	Threshold level should be considered in the context of patient ethnicity
Oligo-anovulation	Oligo-amenorrhea (cycles >35 days apart or <8 menses a year)	If highly suspicious for PCOS, but does not have oligo-amenorrhea, consider serum progesterone or luteinizing hormone assessment
Polycystic ovarian morphology	≥20 follicles per ovary in either ovary ≥10 cm^3^ ovarian volume	Based on transvaginal ultrasonography with a transducer frequency ≥ 8 MHz

Criteria based on the modified 2003 Rotterdam criteria. FAI—free androgen index, BioT—bioavailable testosterone, DHEAS—dehydroepiandrosterone sulfate, ANSD—androstenedione.

## Data Availability

Data are all freely available in pubmed. No new data generated.

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
