# Peer review of "Current Guidelines for Diagnosing PCOS"

_diagnostics, 2023, doi:10.3390/diagnostics13061113_

Round 1

Reviewer 1 Report

I have reviewed the article “Current guidelines for diagnosing PCOS” submitted to Diagnostics. It provides evidence of the evolution of the diagnostic criteria for PCOS over time and a summary of the most recent criteria proposed for the diagnosis of this syndrome. Based on the quality of the article, I recommend publishing it after addressing a mayor revision.

1) Abstract and introduction are too short. It would be better to expand with more details about the methods and main results discussed.

2) The current form of Conclusions is a little simple. It would be better to expand it with more details that are essential.

4) Please double check the whole manuscript for potential grammar errors and typos, especially in the materials and methods section.

5) The English writing needs further proofreading. There are still some linguistic errors and vague descriptions remaining in the manuscript.

Author Response

Reviewer 1: 

I have reviewed the article “Current guidelines for diagnosing PCOS” submitted to Diagnostics. It provides evidence of the evolution of the diagnostic criteria for PCOS over time and a summary of the most recent criteria proposed for the diagnosis of this syndrome. Based on the quality of the article, I recommend publishing it after addressing a mayor revision.

  1. Abstract and introduction are too short. It would be better to expand with more details about the methods and main results discussed.

We have modified the abstract to include a further summary of the methods and main results and conclusions from this review paper. Please see page 1 lines 10-17.

We have added a statement in the introduction to further describe methodology and aims. Please see page 1 line 46 and page 2 lines 47-48.

  1. The current form of Conclusions is a little simple. It would be better to expand it with more details that are essential.

Thank you for this comment. We have reformatted some of our discussions in the conclusion. Given that this is a single chapter in a special issue regarding multiple aspects of PCOS we feel these conclusions fully summarize and comment on our current topic of “current guidelines for diagnosis of PCOS” without overlapping with other invited sections.

  1. Please double check the whole manuscript for potential grammar errors and typos, especially in the materials and methods section.

We have reviewed the entire manuscript and have corrected any grammar errors. There is no materials and methods section in this manuscript.

  1. The English writing needs further proofreading. There are still some linguistic errors and vague descriptions remaining in the manuscript.

Both authors are native English speakers and do not feel any significant linguistic errors are present.

Reviewer 2 Report

The article is an interesting review, it is the time to revise the diagnosis of PCOS

The authors should open new prospectives and should talk about the role of insuline, glucose and microbiota.

Author Response

The article is an interesting review, it is the time to revise the diagnosis of PCOS

The authors should open new prospectives and should talk about the role of insuline, glucose and microbiota.

Thank you for this comment. We agree insulin, glucose, and microbiota are important and interesting aspects of PCOS. However, given that this is a single section in a special issue devoted to multiple aspects of PCOS we feel further discussion of these issues is outside of the current scope of our section of “current guidelines for diagnosis of PCOS.”

Reviewer 3 Report

The manuscript is very well written and is interesting to read.  It summarizes the current guidelines for the diagnosis of PCOS.  I only have three minor comments:

1. The authors should distinguish between the terms "syndrome" and "disease".  PCOS is definitely NOT a disease, therefore this term should NOT be used (page 1, line 28, 33, 37).

2. In the chapter 3.1 entitled "Hyperandrogenaemia":  The authors should be aware that androstenedione and DHEA(-S) cannot be called androgens since they do NOT show any affinity with the androgen receptors (ARs).  They are the adrenal androgen precursors (AAP) which are converted to testosterone or androstenedione and then testosterone, respectively (PMID: 25008465).  Please add 1-2 sentences which will underline this.  The authors should also note that PCOS is actually a diagnosis of exclusion and the measurement of serum testosterone and DHEA-S is always performed in order to exclude ovarian or adrenal androgen secreting tumors, respectively (PMID: 22421986). So, please also comment on that.  I would also suggest to add 1-2 sentences on the differential diagnosis with the NCCAH, which is very common in some populations.  In other words when 17-hydroxyprogesterone should be measured (PMID: 28582566)

3.  In the chapter 4 entitled "Irregular cycles and ovulatory dysfunction" I would suggest to add a comment on the measurement of serum PRL levels in order to exclude hyperprolactinaemia, which is actually a very frequent cause of anovulation (PMID: 22421986)

Author Response

Reviewer 3:

The manuscript is very well written and is interesting to read.  It summarizes the current guidelines for the diagnosis of PCOS.  I only have three minor comments:

  1. The authors should distinguish between the terms "syndrome" and "disease".  PCOS is definitely NOT a disease, therefore this term should NOT be used (page 1, line 28, 33, 37).

Thank you for this comment. We have changed the language throughout to only refer to PCOS as a syndrome.

  1. In the chapter 3.1 entitled "Hyperandrogenaemia":  The authors should be aware that androstenedione and DHEA(-S) cannot be called androgens since they do NOT show any affinity with the androgen receptors (ARs).  They are the adrenal androgen precursors (AAP) which are converted to testosterone or androstenedione and then testosterone, respectively (PMID: 25008465).  Please add 1-2 sentences which will underline this.  

Thank you for this comment. This has now been more specifically defined and clarified. Please see page 3 lines 110-115.

  1. The authors should also note that PCOS is actually a diagnosis of exclusion and the measurement of serum testosterone and DHEA-S is always performed in order to exclude ovarian or adrenal androgen secreting tumors, respectively (PMID: 22421986). So, please also comment on that.  I would also suggest to add 1-2 sentences on the differential diagnosis with the NCCAH, which is very common in some populations.  In other words when 17-hydroxyprogesterone should be measured (PMID: 28582566)

We agree with your comments and have added a section describing the importance of excluding other disorders which may cause hyperandrogenism when evaluating a patient for PCOS. Please see page 3 lines 147-150 and  page 4 lines 151-158 and lines 168-169.

  1.  In the chapter 4 entitled "Irregular cycles and ovulatory dysfunction" I would suggest to add a comment on the measurement of serum PRL levels in order to exclude hyperprolactinaemia, which is actually a very frequent cause of anovulation (PMID: 22421986)

We agree that additional testing to rule out alternative causes of ovulatory dysfunction, including measurement of prolactin levels, should be considered when assessing a patient for PCOS. Please see page 5 lines 229-239.